# Spontaneous Dramatic Regression of Clear Cell Renal Cell Carcinoma After Pazopanib-Induced Severe Systemic Inflammatory Syndrome: A Case Report and Literature Review

**DOI:** 10.3390/curroncol32050260

**Published:** 2025-04-30

**Authors:** Chi Hyuk Oh, Hong Jun Kim

**Affiliations:** 1Department of Internal Medicine, College of Medicine, Kyung Hee University Hospital, 23 Kyung Hee Dae-ro, Dongdaemun-gu, Seoul 02447, Republic of Korea; harrison@daum.net; 2Department of Medical Oncology, College of Medicine, Kyung Hee University Hospital, 23 Kyung Hee Dae-ro, Dongdaemun-gu, Seoul 02447, Republic of Korea

**Keywords:** renal cell carcinoma, spontaneous tumor regression, pazopanib, inflammatory syndrome

## Abstract

Renal cell carcinoma (RCC) is the most common type of kidney cancer, accounting for a significant proportion of all cancer cases in Korea. This case report presents a unique instance of spontaneous dramatic tumor regression in a 42-year-old Korean male diagnosed with clear cell RCC. The patient initially presented with right lower back pain, weight loss, and a loss of appetite. Following systemic immunotherapy with nivolumab and ipilimumab, and right radical nephrectomy, the patient was diagnosed with metastatic clear cell RCC, with new metastatic lesions detected in the liver, and on the chest wall on follow-up imaging. Second-line systemic treatment with pazopanib was initiated. Shortly thereafter, the patient developed severe systemic inflammatory syndrome, resulting in a mental stupor and acute kidney failure. Intensive care, including continuous renal replacement therapy and high-dose immunosuppressants, was administered. The patient’s condition improved significantly with the intensive care regimen, leading to unintended tumor regression. These potentially fatal side effects occurred without infection, as confirmed by negative blood and urine cultures, and were attributed to the recent introduction of pazopanib. Follow-up imaging showed a significant reduction in hepatic metastatic lesions and the disappearance of chest wall nodules. This is the first reported case of RCC tumor regression following the side effects of pazopanib, underscoring the need for further studies into the immune mechanisms involved in RCC treatment and highlighting potential therapeutic strategies that leverage innate immune responses.

## 1. Introduction

Renal cell carcinoma (RCC) is the most common kidney cancer and represents the tenth most common cancer in Korea, accounting for 2.4% of all cancer cases [1]. Its incidence peaks in individuals in their 60s and 70s, and it occurs twice as frequently in males as in females [2]. Over the past 30 years, data from the Korea National Cancer Incidence Database have shown an increase in the incidence of RCC. However, the survival rate has improved [1]. This improvement in the survival rate can be largely attributed to earlier detection facilitated by the widespread use of radiological imaging. Despite these advances, RCC continues to be a fatal malignancy with a crude mortality rate projected of 2.1 per 100,000 individuals in 2023 [3]. It typically presents with a triad of symptoms: flank pain, hematuria, and a palpable abdominal mass. Nonetheless, most patients with RCC are either asymptomatic or present with non-specific symptoms, such as fatigue. Approximately 50% of patients with RCC are discovered incidentally through radiological imaging, with 30% already in advanced stages, including local invasion or metastasis, at the time of diagnosis [4,5].

The spontaneous regression (SR) of tumors is a rare phenomenon; however, it has been observed in various cancers [6,7], particularly in those that can provoke strong immune responses, such as RCC, melanoma, and B-cell malignancies [8]. In RCC, the incidence of SR is approximately 1%, with a higher incidence noted in cases with clear cell histology [9] following resection of the primary tumor [10]. The underlying cause is believed to be an inflammatory response triggered by factors such as drug side effects, severe bacterial or viral infections, and vaccinations, leading to febrile reactions. Since Bumpus (1928) first documented this phenomenon [10], a few cases of spontaneous RCC regression have been recorded and examined, highlighting the significant role of inflammatory responses in these rare events [9]. These immune responses may target and destroy cancer cells, highlighting the potential of the immune system to fight malignancies and suggesting new therapeutic approaches that could mimic this natural process of tumor regression.

Although some cases of spontaneous tumor regression following severe viral or bacterial infections have been reported, instances of cancer regression after potentially fatal side effects of anti-cancer treatments are rare. Herein, we present the case of a 42-year-old male diagnosed with clear cell RCC who experienced tumor regression following the development of systemic inflammatory syndrome shortly after initiating pazopanib therapy. To the best of our knowledge, this is the first reported case of tumor regression following the severe side effects of pazopanib.

## 2. Case Presentation

A 42-year-old Korean man sought evaluation for persistent right-sided lower back discomfort, which he rated as 3 out of 10 on the Numerical Rating Scale and had been experiencing for more than two years without radiation. He had no notable past medical history but reported an unintentional weight loss of 5 kg over the preceding four months along with decreased appetite. His family history included malignancies on his paternal side, with his father having had colorectal cancer and his grandmother uterine cancer. He was a current smoker with a 15 pack-year history and consumed alcohol more than twice weekly. On physical examination, he had pale conjunctiva and a firm, non-tender mass palpable in the right upper abdomen measuring approximately the width of eight fingers. Plain abdominal radiography revealed an area of increased opacity in the right abdomen, raising suspicion for an intra-abdominal mass (Figure 1A). A contrast-enhanced computed tomography (CT) scan showed a large, heterogeneously enhancing renal mass measuring 20 cm involving the right kidney, with extension into the renal pelvis and upper ureter. Tumor thrombus extended into the right renal vein and infra-diaphragmatic inferior vena cava, suggestive of a malignant neoplasm such as RCC staged at T3b or T4 according to the American Joint Committee on Cancer (AJCC) TNM staging system 8th edition (Figure 1B). Histopathology from an ultrasound-guided biopsy confirmed clear cell renal cell carcinoma. As the lesion was deemed unresectable at presentation, tissue confirmation was performed in accordance with NCCN guidelines. To assess for distant disease, a positron emission tomography/computed tomography (PET/CT) scan of the torso was performed and showed a large hypermetabolic lesion confined to the right kidney without distant metastasis (Figure 1C). This imaging was performed to complement the CT findings and facilitate accurate staging for treatment planning.

Due to the large tumor size and extensive thrombosis, a multidisciplinary team recommended neoadjuvant immunotherapy and anticoagulation before surgery. During multidisciplinary discussions, the urology team strongly expressed concerns about the high surgical risk and deemed curative resection unfeasible at that stage due to the tumor’s extensive vascular invasion and associated perioperative complications. As a result, neoadjuvant immunotherapy with nivolumab and ipilimumab was proposed as a second-best option to reduce the tumor size and vascular involvement, with the goal of enabling safer surgical resection at a later time. This strategy was agreed upon by the multidisciplinary team as the most viable approach for this high-risk patient. The patient received 12 cycles of perioperative nivolumab and ipilimumab followed by a right radical nephrectomy, adrenalectomy, and tumor thrombectomy. Macroscopic examination of the surgical specimen showed a large, exophytic, multilobulated tumor with poorly defined margins, measuring 23.3 × 18.0 × 8.5 cm and weighing 3100 g, that had replaced most of the native renal parenchyma. On sectioning, the mass displayed a tan-to-golden yellow appearance, with extensive areas of necrosis and cystic degeneration containing hemorrhagic clots. The tumor extended into the renal sinus fat and infiltrated the wall of the vena cava. Histological analysis confirmed clear cell renal cell carcinoma with sarcomatoid and rhabdoid features, and it was classified as T3b according to the AJCC’s 8th edition, as there was no involvement of the Gerota fascia.

Imaging at a five-month follow-up, revealed new metastatic lesions: small enhancing nodules in the left anterior chest wall (Figure 2A) and a new hepatic metastasis in the segment 7 subcapsular area (Figure 3A). Consequently, second-line systemic treatment with pazopanib at 800 mg, once daily, was initiated. After one week of pazopanib treatment, the patient reported a sore throat and myalgia without fever. His vital signs and liver function test results were normal, except for a mildly elevated C-reactive protein (CRP) level of 4.29 mg/dL, which led to continued monitoring without changes in medication.

On the 10th day of pazopanib therapy, the patient returned with febrile sensations, chills, myalgia, mild cough, sputum, and watery diarrhea. He reported reduced urine output, but no other urinary symptoms. A physical examination revealed a fever of 38.2 °C, pulse rate of 112 bpm, respiratory rate of 20 bpm, and blood pressure of 131/99 mmHg with an alert mental status. Laboratory results revealed elevated CRP (15.72 mg/dL), blood urea nitrogen (BUN) (26 mg/dL), and creatinine (1.96 mg/dL) levels, with a decreased estimated glomerular filtration rate (eGFR) (40 mL/min/1.73 m^2^). These symptoms and findings led to the impression that they were side effects induced by pazopanib, as it was the only new medication introduced prior to the onset of symptoms.

Although the initial impression was that the symptoms were side effects of pazopanib, infection could not be definitively ruled out. Therefore, various infection studies and treatments were concurrently conducted. He was admitted for infection studies and treatment with piperacillin/tazobactam was initiated for presumed infectious enteritis. Despite continued antibiotic treatment, he developed a high fever and significantly elevated CRP (43.03 mg/dL), BUN (61 mg/dL), and creatinine (7.14 mg/dL) levels, with a decrease in urine output and a mental status that worsened into a stupor. He was transferred to the intensive care unit (ICU) where he was intubated and continuous renal replacement therapy (CRRT) was initiated. High-dose methylprednisolone was administered for suspected drug-induced systemic inflammatory syndrome caused by pazopanib. Over three weeks, his renal function gradually recovered, allowing discontinuation of CRRT, and the immunosuppressants were tapered off. His mental status improved to alert, and inflammatory markers, including CRP, normalized. Blood and urine cultures did not reveal any evidence of infection, which led to the discontinuation of empirical antibiotics. By the time of discharge, his eGFR had improved to 16 mL/min/1.73 m^2^, and it further increased to 63 mL/min/1.73 m^2^ over the next four months. The patient returned to daily activities, and follow-up imaging performed approximately four months after the onset of systemic inflammatory syndrome revealed that the chest wall nodules had disappeared (Figure 2B) and the hepatic lesion had significantly decreased in size (Figure 3B). According to RECIST 1.1 criteria, the disappearance of the chest wall nodules (non-target lesions) constituted a complete response, while the hepatic lesion demonstrated over 90% shrinkage without evident viable tissue, consistent with a partial response (PR). Therefore, the overall tumor response was categorized as PR. The patient has remained progression-free for 16 months after recovery, without additional anticancer therapy. The most recent follow-up imaging was performed in February 2025, and showed no evidence of disease progression.

## 3. Discussion

In this case, a 42-year-old male diagnosed with clear cell RCC underwent successful systemic immunotherapy with 12 cycles of nivolumab and ipilimumab, followed by right radical nephrectomy. Five months later, metastatic lesions were detected in the liver and chest wall, prompting initiation of pazopanib treatment. Shortly after commencing pazopanib, the patient developed a systemic inflammatory syndrome, leading to a mental stupor and kidney failure, necessitating his admission to the ICU, intubation, CCRT, and immuno-suppressive treatment. Remarkably, unintended tumor regression was observed during the patient’s recovery. The treatment approach, including the use of nivolumab and ipilimumab, was consistent with NCCN guidelines, which recommend first-line systemic therapy for unresectable, advanced RCC. This case is significant as it diverges from the commonly reported SR in RCC, which typically manifests as regression of metastatic lesions following the removal of the primary tumor [10]. In contrast, this case involved a relapse of the cancer five months after primary tumor resection and subsequent tumor regression induced by the severe side effects of an anticancer agent. Although tumor regression following severe viral or bacterial infections is relatively well-documented [11,12,13], reports of regression after the potentially fatal side effects of anticancer drugs are exceedingly rare. The patient’s condition worsened despite initial empirical antibiotic treatment, and no evidence of infection was found in blood or urine cultures, ruling out an infectious cause. Additional diagnostic workup, including serum procalcitonin levels, fungal antigen tests (G and GM), and viral serologies for hepatitis B, hepatitis C, CMV, and EBV, also yielded negative results. Autoimmune markers, such as ANA and ANCA, were within normal limits, and no clinical signs of autoimmune disease were noted. Furthermore, no other new medications or environmental exposures were identified. Given the short duration of pazopanib use and the established response rate data [14], it is unlikely that the observed tumor regression was due to a direct effect of the anticancer drug. Pazopanib was the only new medication introduced before symptom onset. The patient showed significant improvement with sufficient supportive care, including prolonged CRRT and the administration of immunosuppressants. These findings suggest that the severe systemic inflammatory response was a pazopanib-induced drug reaction and played a pivotal role in the unintended regression of the metastatic RCC.

Renal cell carcinoma accounts for approximately 3% of all cancers [15]. The prognosis of patients with RCC is closely linked to tumor stage and cytologic characteristics [16]. The TNM system from the American Joint Committee on Cancer staging manual, 8th edition, is commonly used to stage tumors [17]. Approximately 65% of patients with RCC are present with localized tumors, 16% have regional lymph node involvement, and 16% present with distant metastases [18]. Imaging, particularly contrast-enhanced CT, is crucial for detecting and staging RCC, although it has limitations in determining the invasion of important anatomical landmarks such as the renal capsule or perirenal fascia [19]. Surgical resection is the standard treatment for localized RCC, and often results in favorable outcomes [20]. Patients with low-risk tumors typically undergo partial or radical nephrectomy. In cases of high-risk localized tumors and those with metastatic disease with a good-to-intermediate prognosis, a combination of surgery and systemic treatments may be used. Systemic treatments include targeted therapies and immunotherapies [20]. Before 2006, the treatment options were limited to cytokine therapies, such as interferon-alpha and interleukin-2, which had limited efficacy and high toxicity [21,22]. The advent of tyrosine kinase inhibitors between 2006 and 2010 significantly improved the median survival time from approximately 15 to 30 months [23]. More recently, immune checkpoint inhibitors, such as nivolumab, ipilimumab, and pembrolizumab, either alone or in combination, have shown considerable efficacy [24,25]. Historically, even in an era when cytotoxic chemotherapy was the standard treatment for solid tumors, cytokine therapy demonstrated durable or complete remission in a small group of patients [21,22]. This, together with the recently reported high efficacy of immune checkpoint inhibitors [24,25], underscores the potential impact of immune reactions in the treatment of RCC. The established efficacy of these inhibitors suggests that harnessing immune responses is a crucial strategy for managing RCC. This case, in which drug-induced heightened innate immunity led to cancer regression, highlights the importance of exploring immune-mediated therapeutic approaches, particularly given their demonstrated benefits in improving the outcomes of patients with RCC.

Spontaneous regression of cancer is a rare yet well-recognized phenomenon characterized by the partial or complete disappearance of a tumor without any treatment capable of causing regression [6]. Several mechanisms have been suggested to explain SR, including immune or hormonal mediation, tumor inhibition by growth factors or cytokines, in-duction of differentiation, elimination of carcinogens, tumor necrosis, inhibition of angio-genesis, psychological factors, apoptosis, and epigenetic modifications [26]. A leading hypothesis posits that the host immune response to a pathogen, whether a microorganism, foreign substance, or drug, plays a crucial role in cancer regression [27]. This immune response is intricate and not fully understood, often involving a phase of hyperactive inflammation, followed by anti-inflammatory responses [27]. Fever is commonly observed in hyperactive immune states and can boost cytokine secretion by immune cells and elevate proinflammatory cytokine levels [28]. Moreover, cancer cells are more vulnerable than normal cells to heat-induced apoptosis. Necrotic or heat-stressed cancer cells can serve as antigens or pathogen-associated molecular patterns, thereby creating an immune-stimulating environment [28]. For instance, in patients with metastatic melanoma undergoing immunotherapy, a fever of 39.5 °C or higher has been recognized as an independent factor for better survival and tumor response [29]. In the context of RCC, SR is most commonly associated with nephrectomy, which is the standard treatment [30]. This surgical procedure may facilitate regression by eliminating the primary source of tumor-promoting substances and allowing the immune system to target residual cancer cells more effectively [31]. Removal of the primary tumor reduces the burden of cancer antigens and may heighten antigen exposure during the dissemination of tumor cells, potentially provoking a strong anti-tumor immune response [32]. Thus, the immune mechanism is strongly implicated as a significant factor in SR of RCC.

Pazopanib is an oral multi-kinase inhibitor targeting a range of receptors, including vascular endothelial growth factor receptors-1, -2, and -3, platelet endothelial growth fac-tor receptors-α and -β, interleukin-2 receptor-inducible T-cell kinase, leukocyte-specific protein tyrosine kinase, colony-stimulating factor-1 receptor, fibroblast growth factor re-ceptors-1 and -3, and c-kit [33]. Although initially developed for various cancers, pazopanib is currently approved for the treatment of advanced soft-tissue sarcoma and RCC. A pivotal phase III trial demonstrated that pazopanib significantly prolonged progression-free survival (PFS) compared to the placebo in treatment-naïve patients with RCC [34], whereas a phase II trial showed a 27% objective response rate, a 49% stable disease rate, and a median PFS of 7.5 months with a 24-month overall survival rate of 43% in patients previously treated with other agents [35]. Common side effects of pazopanib include hypertension (40.1%), fatigue (36.6%), hypopigmentation (36.6%), nausea (35.9%), and diarrhea (30.3%). Additionally, patients may experience increased liver enzyme levels, myelosuppression, and proteinuria, mostly Grade 1 or 2 according to the National Cancer Institute Common Terminology Criteria for Adverse Events (CTCAE) v3.0. Grade 3 or higher side effects primarily included hypertension (7.7%), fatigue (7.7%), and hyperbilirubinemia (6.3%) [36]. Treatment interruption occurred in 60% of cases, with 23% requiring dose reduction, reflecting the drug’s overall tolerability. Severe toxicities leading to treatment discontinuation, including bowel perforation and pulmonary embolism, were observed in 6% of cases [36]. Pazopanib has also been associated with acute kidney injury (AKI), as highlighted in prior studies, which have attributed this to vascular endothelial growth factor receptor inhibition causing endothelial and podocyte injury, as well as microangiopathy [37]. In the present case, the observed AKI and other systemic symptoms align with previously reported side effects of pazopanib. However, the severe systemic inflammatory syndrome observed in this patient may have amplified the immune-mediated mechanisms, contributing to the unexpected tumor regression. This underscores the duality of pazopanib’s impact as both a therapeutic and a modulator of immune responses. Recent studies have proposed that VEGFR inhibitors, including pazopanib, can trigger immune-related pathways beyond their anti-angiogenic effects [38]. These include an activation of the STING (stimulator of interferon genes) pathway and the upregulation of type I interferon responses, potentially enhancing innate immune activation. Such mechanisms may have synergized with the patient’s systemic inflammatory state, promoting immune-mediated tumor regression. In this case, the treatment approach aligned with the current guidelines for managing advanced RCC, incorporating immune checkpoint inhibitors followed by targeted therapies [39]. Despite the known side effects of pazopanib, there have been no documented cases of tumor regression following its discontinuation due to side effects, making this case the first reported instance of spontaneous tumor regression under these circumstances.

## 4. Conclusions

This case represents the first documented instance of tumor regression in a patient with RCC following the severe side effects of pazopanib. The uniqueness of this case lies in the fact that SR did not occur in association with an infection or following primary tumor resection, which are the typically reported triggers. This highlights the need for further studies into the mechanisms by which RCC may be specifically susceptible to innate immune responses. A deeper understanding of these mechanisms could provide significant insights and assist in the development of more effective treatment strategies for RCC.

## Figures and Tables

**Figure 1 curroncol-32-00260-f001:**
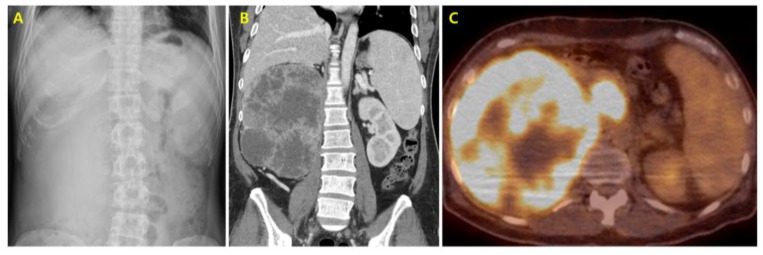
(**A**) Erect view of abdominal radiography showing a mass-like opacity in the right abdomen. (**B**) Contrast-enhanced abdominal computed tomography revealing a heterogeneous enhancing mass occupying the right kidney, with nephromegaly measuring 19.1 cm in length. The mass extends into the pelvicalyceal system and proximal ureter and is associated with tumor thrombosis in the right renal vein, infra-diaphragmatic inferior vena cava, and at the branching site of the left renal vein. (**C**) Axial view of positron emission tomography/computed tomography (PET/CT) of the torso, showing a large hypermetabolic lesion in the right kidney with a maximum standardized uptake value (SUV) of 13.3 (consistent with renal cell carcinoma and involving the proximal ureter.

**Figure 2 curroncol-32-00260-f002:**
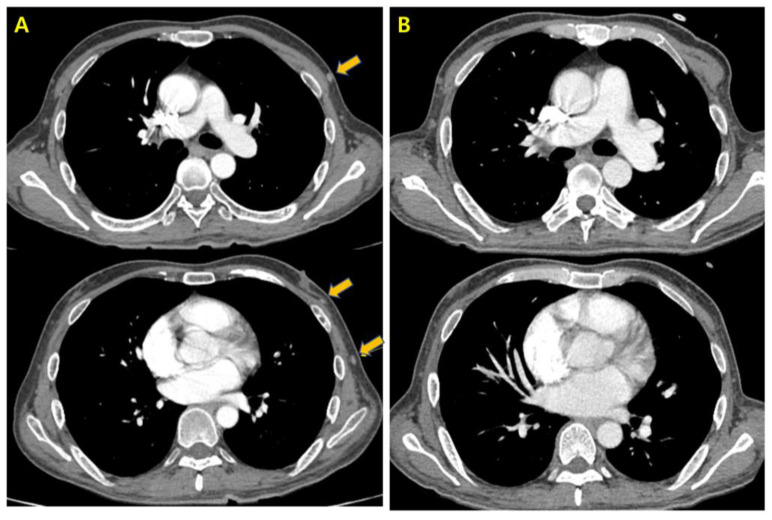
(**A**) Contrast-enhanced chest computed tomography revealing the appearance of new and gradually enlarging small nodules (yellow arrows) in the left anterior chest wall, suggestive of metastasis. (**B**) Follow-up chest computed tomography demonstrating the disappearance of these nodules.

**Figure 3 curroncol-32-00260-f003:**
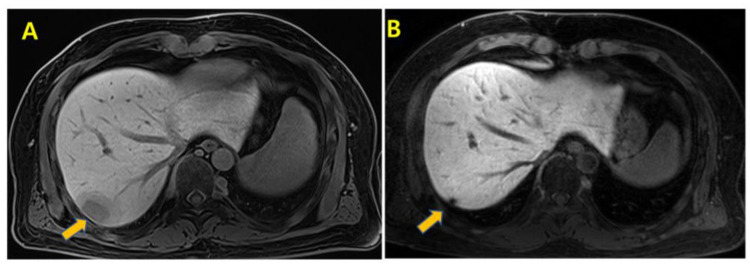
(**A**) Dynamic contrast-enhanced magnetic resonance imaging (MRI) of the liver showing a rim-enhancing mass (yellow arrow) in the subcapsular area of segment 7 (S7) with peritumoral hyperemic and edematous parenchymal changes, diffusion restriction, and internal linear hypointense foci on T1-weighted and T2-weighted images with signal drop at the in-phase sequence. (**B**) Follow-up MRI revealing a decreased remnant metastatic lesion (yellow arrow) in the posterior margin of S7 with uncertain viability.

## Data Availability

Data are contained within the article.

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
