# Peer review of "Spontaneous Dramatic Regression of Clear Cell Renal Cell Carcinoma After Pazopanib-Induced Severe Systemic Inflammatory Syndrome: A Case Report and Literature Review"

_curroncol, 2025, doi:10.3390/curroncol32050260_

Round 1
Reviewer 1 Report
Comments and Suggestions for Authors
This case report describes a rare phenomenon of spontaneous regression of metastatic clear cell renal cell carcinoma (cRCC) following pazopanib-induced severe systemic inflammatory syndrome, representing the first reported case of its kind. The study provides a detailed clinical course and imaging evidence, highlighting a potential link between the toxic effects of targeted therapy and antitumor immune responses, which offers a novel perspective on the "toxicity-efficacy conversion" in renal cancer treatment.
Comments
1. Enhance Follow-up Data Details:
The manuscript states the patient remained progression-free for "over one year." Please specify the exact follow-up duration (e.g., 18 months) and the date of the most recent imaging assessment to strengthen the credibility of the results.
2. Strengthen the Description of Differential Diagnosis for Inflammatory Syndrome:
While the discussion notes an "infectious etiology was ruled out," elaborate on specific diagnostic tests performed (e.g., procalcitonin, G/GM tests, viral serology) to fully demonstrate the infection workup and address potential reader concerns about non-infectious inflammation. Include a brief analysis of other possible causes of systemic inflammation (e.g., autoimmune diseases, adverse reactions other than drug allergy) to reinforce the conclusion that pazopanib was the sole trigger through differential diagnosis.
Clarify Imaging Assessment Criteria (Result Visualization):
3. In Figures 2 and 3, where chest wall nodules "disappeared" and hepatic metastases "significantly reduced," specify the imaging evaluation criteria (e.g., RECIST 1.1 criteria for target lesion diameter changes and response classification) to standardize results for comparability.
State the interval between the two imaging examinations in the "Case Presentation" or "Discussion" to clarify the temporal association between tumor regression and inflammatory syndrome treatment.
4. Deepen Mechanistic Discussion with Additional Literature:
While the discussion touches on innate immune responses and cytokines, incorporate recent research on "targeted drug-induced immune activation" (e.g., VEGFR inhibitors associated with STING pathway activation or type I interferon responses) to enhance the depth of mechanistic analysis.
5. In the abstract, "innate immune responses" is appropriate, but consider adding "inflammatory syndrome" or "drug-induced toxicity" to the keywords (currently "side effects" is too broad) for better alignment with the study’s core focus.
Author Response
Comments
- Enhance Follow-up Data Details:
The manuscript states the patient remained progression-free for "over one year." Please specify the exact follow-up duration (e.g., 18 months) and the date of the most recent imaging assessment to strengthen the credibility of the results.
Response 1:
Thank you for this insightful comment. We agree that providing a more specific follow-up duration and the timing of the most recent imaging enhances the transparency and accuracy of the report. Accordingly, we have revised the relevant sentence in the Case Presentation section, final paragraph (Page 4, Paragraph 1, line 156) as follows:
“[The patient has remained progression-free for 16 months after recovery, without additional anticancer therapy. The most recent follow-up imaging was performed in February 2025, and showed no evidence of disease progression.]”
[The patient has remained progression-free for 16 months after recovery, over one year without additional anticancer therapy. The most recent follow-up imaging was performed in February 2025, and showed no evidence of disease progression.]
This revision clarifies the follow-up period and supports the validity of the observed durable response.
Strengthen the Description of Differential Diagnosis for Inflammatory Syndrome:
While the discussion notes an "infectious etiology was ruled out," elaborate on specific diagnostic tests performed (e.g., procalcitonin, G/GM tests, viral serology) to fully demonstrate the infection workup and address potential reader concerns about non-infectious inflammation. Include a brief analysis of other possible causes of systemic inflammation (e.g., autoimmune diseases, adverse reactions other than drug allergy) to reinforce the conclusion that pazopanib was the sole trigger through differential diagnosis.
Clarify Imaging Assessment Criteria (Result Visualization):
Response 2:
Thank you for this insightful and important comment. We agree that a more detailed explanation of the infection workup and differential diagnosis of the systemic inflammatory syndrome would strengthen the credibility of our interpretation. We have now revised the Discussion section (Page 5, Paragraph 1, line 189) to include the specific tests performed during the infection workup and to acknowledge other possible causes of systemic inflammation. We also clarified that no autoimmune-related findings were observed and emphasized the exclusion of other medications or recent exposures that could have contributed.
We believe the request to “Clarify Imaging Assessment Criteria (Result Visualization)” more directly relates to Comment 3, which we have addressed in detail in our response there. Thank you again for your thoughtful feedback.
Revised text (Discussion section):
The patient's condition worsened despite initial empirical antibiotic treatment, and no evidence of infection was found in blood or urine cultures, ruling out an infectious cause. Additional diagnostic workup, including serum procalcitonin levels, fungal antigen tests (G and GM), and viral serologies for hepatitis B, hepatitis C, CMV, and EBV, also yielded negative results. Autoimmune markers, such as ANA and ANCA, were within normal limits, and no clinical signs of autoimmune disease were noted. Furthermore, no other new medications or environmental exposures were identified. Given the short duration of pazopanib use and the established response rate data [14], it is unlikely that the observed tumor regression was due to a direct effect of the anticancer drug. Pazopanib was the only new medication introduced before symptom onset. The patient showed significant improvement with sufficient supportive care, including prolonged CRRT and the administration of immunosuppressants. These findings suggest that the severe systemic inflammatory response was a pazopanib-induced drug reaction and played a pivotal role in the unintended regression of the metastatic RCC.
Clarify Imaging Assessment Criteria (Result Visualization):
In Figures 2 and 3, where chest wall nodules "disappeared" and hepatic metastases "significantly reduced," specify the imaging evaluation criteria (e.g., RECIST 1.1 criteria for target lesion diameter changes and response classification) to standardize results for comparability.
State the interval between the two imaging examinations in the "Case Presentation" or "Discussion" to clarify the temporal association between tumor regression and inflammatory syndrome treatment.
Response 3:
Thank you for this valuable suggestion. We agree that clearly stating the imaging response assessment criteria improves the interpretability of the manuscript. We have clarified that the evaluation was based on RECIST 1.1 criteria, and provided classifications accordingly: the chest wall nodules, as non-target lesions, were classified as complete response (CR) due to complete disappearance on follow-up imaging. The hepatic lesion, a target lesion, demonstrated over 90% reduction in size and no apparent viable tumor tissue, thus qualifying as a near-complete response but conservatively categorized as partial response (PR).
We have also specified that the interval between the two imaging assessments was approximately 4 months, aligning with the patient’s recovery period from the systemic inflammatory syndrome.
Revised Text (Page 4, Case Presentation, Final Paragraph, line 149):
“The patient returned to daily activities, and follow-up imaging performed approximately four months after the onset of systemic inflammatory syndrome revealed showed that the chest wall nodules had disappeared (Fig 2B) and the hepatic lesion had significantly decreased in size there was a significant reduction in the size of the metastatic hepatic lesions (Fig 3B). According to RECIST 1.1 criteria, the disappearance of the chest wall nodules (non-target lesions) constituted a complete response, while the hepatic lesion demonstrated over 90% shrinkage without evident viable tissue, consistent with a partial response (PR). Therefore, the overall tumor response was categorized as PR.”
Deepen Mechanistic Discussion with Additional Literature:
While the discussion touches on innate immune responses and cytokines, incorporate recent research on "targeted drug-induced immune activation" (e.g., VEGFR inhibitors associated with STING pathway activation or type I interferon responses) to enhance the depth of mechanistic analysis.
Response 4:
Thank you for this insightful comment. We agree that expanding on the mechanistic basis of pazopanib-induced immune activation would enrich the discussion and align with recent findings in the field. Accordingly, we have revised the final paragraph of the Discussion section to incorporate literature regarding VEGFR inhibitor-mediated activation of the STING pathway and subsequent type I interferon signaling. This addition provides a more comprehensive explanation of how targeted agents such as pazopanib may modulate innate immunity beyond their canonical anti-angiogenic effects.
I also appropriately added to the revised reference list as Reference 38.
Revised Text (Discussion, Final Paragraph, page 7, line 282):
In the present case, the observed AKI and other systemic symptoms align with previously reported side effects of pazopanib. However, the severe systemic inflammatory syndrome observed in this patient may have amplified the immune-mediated mechanisms, contributing to the unexpected tumor regression. This underscores the duality of pazopanib's impact as both a therapeutic and a modulator of immune responses. Recent studies have proposed that VEGFR inhibitors, including pazopanib, can trigger immune-related pathways beyond their anti-angiogenic effects [38]. These include activation of the STING (stimulator of interferon genes) pathway and upregulation of type I interferon responses, potentially enhancing innate immune activation. Such mechanisms may have synergized with the patient’s systemic inflammatory state, promoting immune-mediated tumor regression. In this case, the treatment approach aligned with the current guidelines for managing advanced RCC, incorporating immune checkpoint inhibitors followed by targeted therapies [39]. Despite the known side effects of pazopanib, there have been no documented cases of tumor regression following its discontinuation due to side effects, making this case the first reported instance of spontaneous tumor regression under these circumstances.
In the abstract, "innate immune responses" is appropriate, but consider adding "inflammatory syndrome" or "drug-induced toxicity" to the keywords (currently "side effects" is too broad) for better alignment with the study’s core focus.
Response 5:
Thank you for this helpful suggestion. We agree that "side effects" is a broad term and that incorporating more specific keywords would better reflect the manuscript’s focus. Therefore, we have replaced "side effects" with “inflammatory syndrome” in the keyword list to improve alignment with the content.
Revised Text (Page 1, Abstract - Keywords):
“Keywords: renal cell carcinoma; spontaneous tumor regression; pazopanib; inflammatory syndrome”
Reviewer 2 Report
Comments and Suggestions for Authors
This is an interesting case report. However, the authors may need to consider the immune reconstitution inflammatory syndrome (IRIS) and discuss this possible mechanism. The comprehensive search for chronic infection such as viral infection or parasitologic infection in this kind of patients was needed before the immunotherapy.
The authors need to change «unrespectable» in line 172 to «unresectable»?
Author Response
This is an interesting case report. However, the authors may need to consider the immune reconstitution inflammatory syndrome (IRIS) and discuss this possible mechanism.
The comprehensive search for chronic infection such as viral infection or parasitologic infection in this kind of patients was needed before the immunotherapy.
Response
We sincerely thank the reviewer for this insightful and thought-provoking comment. We had not previously considered the possibility of immune reconstitution inflammatory syndrome (IRIS) in the context of this case, and we genuinely appreciate the opportunity to reflect on this important immunologic mechanism. If IRIS were indeed applicable, it could add an additional layer of interest and mechanistic depth to this already unusual clinical course.
However, after careful re-evaluation, we believe that this case may not fully align with the typical clinical context and criteria for IRIS. We say this with some regret, as we acknowledge the value of the reviewer’s perspective and wish it could apply more directly. Below, we respectfully outline our reasoning:
- No inflammatory complications occurred following immunotherapy: The patient received 12 cycles of perioperative nivolumab and ipilimumab without experiencing any signs or symptoms of systemic inflammation throughout or following the treatment period. Prior to immunotherapy initiation, routine viral screening (e.g., hepatitis B and C, HIV) revealed no abnormalities, and there were no clinical indicators suggestive of latent or chronic infection.
- The inflammatory syndrome occurred after the initiation of pazopanib, not during a period of immune recovery. At that time, we conducted a thorough infectious work-up, including blood and urine cultures, procalcitonin levels, viral serologies (hepatitis B/C, CMV, EBV), and fungal markers (G/GM), all of which were negative. Autoimmune causes were also considered and excluded. We realize in hindsight that the manuscript initially underemphasized the extent of this evaluation, and we sincerely apologize if this led to any confusion or misinterpretation. We have since revised the discussion section to more clearly describe the comprehensive infection workup performed during the acute illness: The severe systemic inflammatory syndrome developed only after the patient’s disease had relapsed and pazopanib was initiated. As described in our revised manuscript (Page XX, Discussion section), we conducted an extensive infectious disease workup at that time, including serial blood and urine cultures, procalcitonin, fungal antigen (G/GM) testing, viral serologies (e.g., HBV, HCV, CMV, EBV), and autoimmune markers. All results were negative. These findings support our conclusion that the inflammatory syndrome was most consistent with a pazopanib-induced drug reaction, rather than IRIS.
Revised text (Discussion section Page 5, Paragraph 1, line 189):
The patient's condition worsened despite initial empirical antibiotic treatment, and no evidence of infection was found in blood or urine cultures, ruling out an infectious cause. Additional diagnostic workup, including serum procalcitonin levels, fungal antigen tests (G and GM), and viral serologies for hepatitis B, hepatitis C, CMV, and EBV, also yielded negative results. Autoimmune markers, such as ANA and ANCA, were within normal limits, and no clinical signs of autoimmune disease were noted. Furthermore, no other new medications or environmental exposures were identified. Given the short duration of pazopanib use and the established response rate data [14], it is unlikely that the observed tumor regression was due to a direct effect of the anticancer drug. Pazopanib was the only new medication introduced before symptom onset. The patient showed significant improvement with sufficient supportive care, including prolonged CRRT and the administration of immunosuppressants. These findings suggest that the severe systemic inflammatory response was a pazopanib-induced drug reaction and played a pivotal role in the unintended regression of the metastatic RCC.
- The patient was not immunocompromised prior to the event: IRIS is typically associated with immune recovery in patients who were previously immunosuppressed, most notably individuals with HIV beginning antiretroviral therapy. In contrast, our patient did not have any underlying immunodeficiency and was not receiving immunosuppressive therapy at the time of the inflammatory event. Therefore, the immunologic context does not support IRIS as a likely diagnosis.
Once again, we deeply appreciate the reviewer’s thoughtful suggestion, which led us to enrich the discussion and clarify key diagnostic considerations. Thank you for helping to strengthen the manuscript.
The authors need to change «unrespectable» in line 172 to «unresectable»?
Response 1:
Thank you for identifying this typographical error. We agree with your correction. The term “unrespectable” has been revised to “unresectable” to accurately reflect the intended clinical meaning. (Page 5, line 182)